# Evolutionary history of zoogeographical regions surrounding the Tibetan Plateau

Jiekun He [1], Siliang Lin [1], Jiatang Li[2], Jiehua Yu[1] & Haisheng Jiang[1 ✉]

The Tibetan Plateau (TP) and surrounding regions have one of the most complex biotas on Earth. However, the evolutionary history of these regions in deep time is poorly understood. Here, we quantify the temporal changes in beta dissimilarities among zoogeographical regions during the Cenozoic using 4,966 extant terrestrial vertebrates and 1,278 extinct mammal genera. We identify ten present-day zoogeographical regions and find that they underwent a striking change over time. Specifically, the fauna on the TP was close to the Oriental realm in deep time but became more similar to the Palearctic realms more recently. The present-day zoogeographical regions generally emerged during the Miocene/Pliocene boundary (*ca.* 5 Ma). These results indicate that geological events such as the Indo-Asian Collision, the TP uplift, and the aridification of the Asian interior underpinned the evolutionary history of the zoogeographical regions surrounding the TP over different time periods.

[1] Spatial Ecology Lab, School of Life Sciences, South China Normal University, 510631 Guangzhou, China. [2] Chengdu Institute of Biology, Chinese Academy of Sciences, 610041 Chengdu, China. ✉email: jhs@scnu.edu.cn

The Tibetan Plateau (TP) uplift was one of the most important geological events in the Cenozoic era (~65 Ma–present[1,2]). It substantially modified the topography[3] and atmospheric circulation[4] of Asia (Fig. 1) and resulted in one of the most complex biotas on Earth[5]. Eight major zoogeographical regions were recently identified surrounding the TP[5,6], namely, the Mongolian Plateau, Central Asia, North Asia, West Asia, South Asia, Southeast Asia, South China and North China (Fig. 1). However, it is unknown how these present-day zoogeographical regions evolved over geological time, even though this information is crucial for understanding the origin and evolution of life in Asia.

There is growing evidence that the present-day zoogeographical regions surrounding the TP are the products of geological processes and past climatic changes[7–9]. A common hypothesis is that the TP uplift created species-dispersal barriers during the Cenozoic[10,11], and subsequent climatic changes in Asia increased environmental heterogeneity (Fig. 1, see refs. [12,13]); also both events caused the geographical isolation of resident lineages and facilitated the differentiation of zoogeographical regions[14]. Other biogeographical analyses, however, revealed that historical events, such as the Eocene Indo-Asian Collision[15,16], the intercontinental biotic exchange between Eurasia and North America[17] and the Pleistocene glaciation cycle[18], might have expanded species' ranges, promoted dispersal and attenuated the faunistic dissimilarities among regions. These processes have been proved to facilitate dispersal and vicariance for many lineages that might increase or decrease the number of taxa common to different regions[19], and ultimately alter their pairwise faunistic relationships over time. However, it remains uncertain how these processes shaped the evolution and emergence of the present-day zoogeographical regions surrounding the TP.

Recent phylogeographical analyses on the TP have associated biogeographical and evolutionary lineage relationships with specific geological events and periods[14]. Unfortunately, most available empirical studies have relied upon the interpretation of single-taxon analyses. They have inferred the influences of geological processes and climatic shifts on genus- or species-level distributions of specific taxa[10,11,13]. However, responses to common geological events might greatly vary among lineages owing to their biological and ecological differences[20]. These differences would result in incongruent biogeographical patterns across different taxonomic lineages over space or time[7,14]. Furthermore, present-day zoogeographical regions were structured by a combination of multiple speciations, extinctions and dispersal processes at several time periods[21,22]. Thus, biogeographical meta-analysis[16,22] and community-level analyses[5,23], which integrate individual taxon histories into shared biotic area histories, were more promising to clarify the processes shaping biogeographical regions over time[19,24].

To date, two primary analyses of community-level data have been used to reconstruct the evolutionary history of zoogeographical regions. One tracked temporal changes in beta diversity between extant communities over a phylogenetic timescale[23,24], and another compared compositional dissimilarities among fossil assemblages over geological time[9,25]. However, both methods have their pros and cons[26]. The former method provides a finer resolution regarding the spatial and temporal changes in communities[23], but always fails to deal with past extinctions and distribution changes[27] and, therefore, provides only indirect evidence. Although the inclusion of ancestral range reconstruction in quantifying phylogenetic dissimilarity can improve estimates of evolutionary history, it is difficult to incorporate extinct lineages into the analysis (ref. [24], but see refs. [28,29]). In contrast, palaeontological materials can provide a direct record of past changes in communities, but they always suffer from incomplete

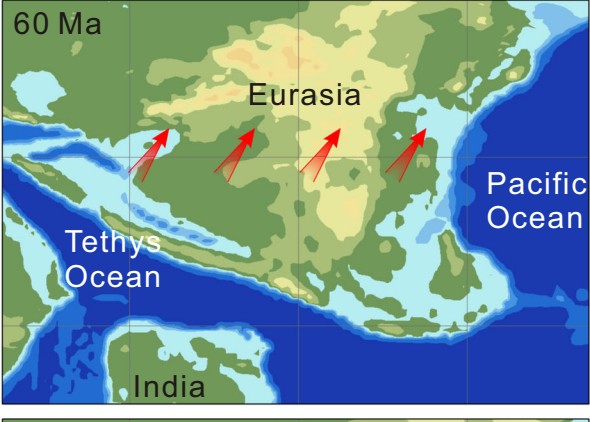

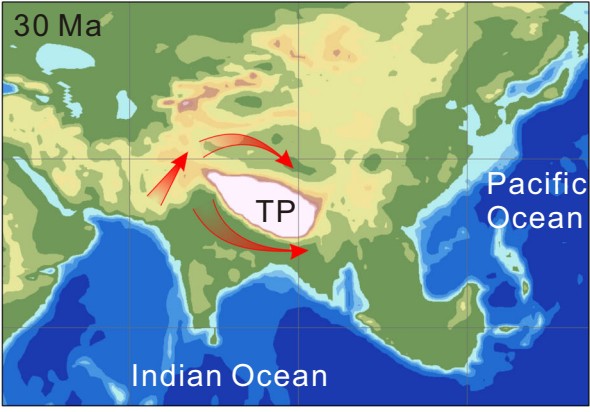

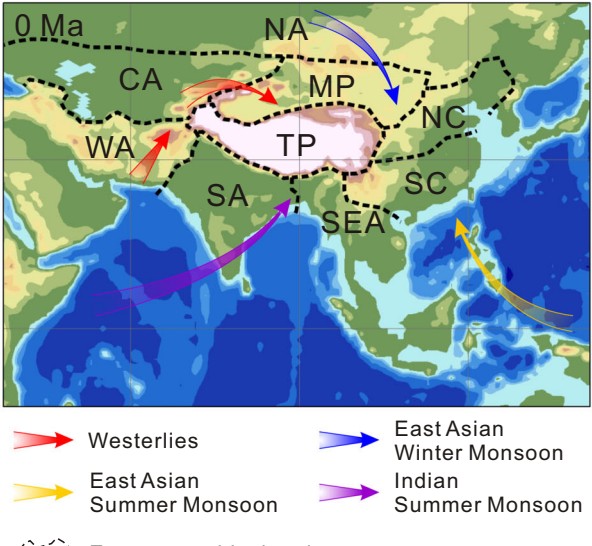

**Fig. 1 Palaeogeographical processes in the Tibetan Plateau and surrounding regions during the Cenozoic.** Palaeotopography was derived from a palaeo-digital elevation model (palaeoDEM, 1° × 1° resolution) developed by Scotese & Wright[52]. Changes in the atmosphere–ocean climate system were compiled from the data of Sun & Wang[50]. Present-day zoogeographical regions were adapted from Kreft & Jetz[6] and Holt et al.[5]. CA Central Asia, MP Mongolian Plateau, NA North Asia, NC North China, SA South Asia, SC South China, SEA Southeast Asia, TP Tibetan Plateau, WA West Asia.

preservation[30], which possibly conceals some important signs of biogeographical events[31]. Nevertheless, despite the limitations of the respective methods, phylogenetic and palaeontological analyses can usefully complement each other in biogeographical

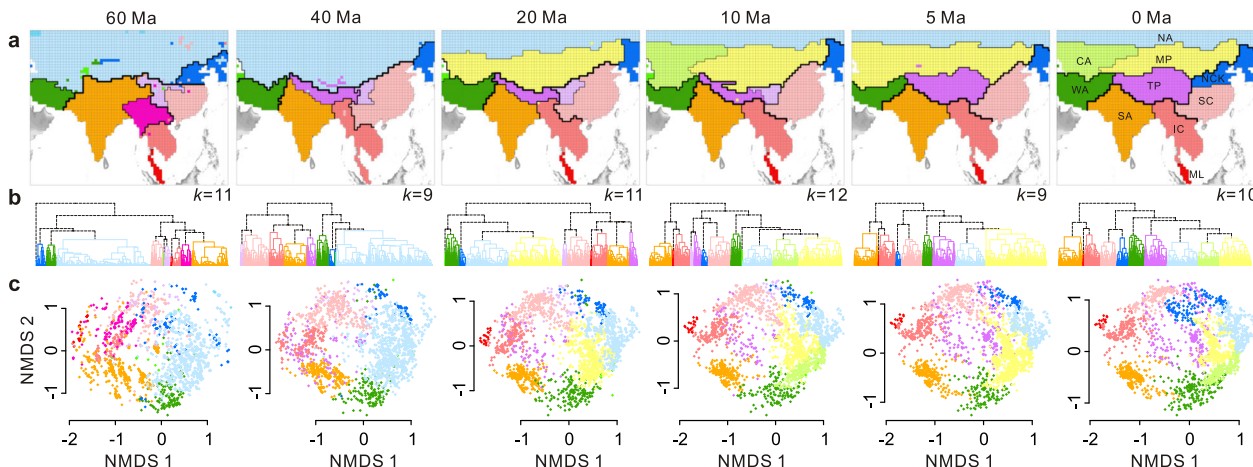

**Fig. 2 Temporal changes in the zoogeographical regions surrounding the Tibetan Plateau at successive phylogenetic depths during the Cenozoic.**
**a** Map showing zoogeographical regions based on pβ$_{sim}$ dissimilarity between pairs of grid-based terrestrial vertebrate communities at different phylogenetic depths. The width of the boundary was scaled to the β$_{sim}$ dissimilarity, with thinner lines showing lower β$_{sim}$ dissimilarities. CA Central Asia, IC Indochina, ML Malay Peninsula, MP Mongolian Plateau, NA North Asia, NCK North China & Korea, SA South Asia, SC South China, TP Tibetan Plateau, WA West Asia. **b** Dendrograms plotted by the unweighted pair-group method using arithmetic average clustering. **c** Coordinates for non-metric multidimensional scaling ordination based on the pβ$_{sim}$ dissimilarity matrix for grid-based terrestrial vertebrate communities.

studies[26], and their correspondence has begun to emerge in large-scale biogeographical contexts[32,33].

In this study, we reconstructed the evolutionary history of the zoogeographical regions surrounding the TP using 4966 extant terrestrial vertebrates along a phylogenetic timescale and 1278 extinct mammal genera over geological time. By comparing analyses implemented over phylogenetic and geological time-scales, we aimed to explore the timeframe within which the present-day zoogeographical regions evolved during the Cenozoic era and estimate the time when the present-day spatial structure of the zoogeographical regions emerged. To reconstruct historical changes in the zoogeographical regions, we quantified the phylogenetic beta dissimilarity using extant species along the phylogenetic timescale. For extinct lineages, we calculated beta dissimilarity based on mammal fossil assemblages over geological time. We assessed the changes in assignments and topologies of hierarchical clustering dendrograms and relative positions in ordinations based on beta dissimilarity at different phylogenetic depths and geological periods. Finally, we explored the relationships between the evolutionary history of the zoogeographical regions within the context of geological and climatic events. Our study reveals that the zoogeographical regions underwent a striking change during the Cenozoic era, and broadly emerged in the Miocene/Pliocene boundary (ca. 5 Ma) owing to a series of geological events such as the Indo-Asian Collision, the TP uplift and the aridification of the Asian interior.

## Results
**Zoogeographical regions over phylogenetic time.** Ten present-day zoogeographical regions were delineated by unweighted pair-group method using arithmetic average (UPGMA) clustering based on pβ$_{sim}$ matrix, namely the Tibetan Plateau, Mongolian Plateau, Central Asia, North Asia, West Asia, South Asia, Indochina, Malay Peninsula, South China and North China & Korea (Fig. 2). At a phylogenetic depth of 60 Ma, nine clustered zoogeographical regions were identified (Fig. 2). They roughly corresponded to the Palearctic realm, North China & Korea, West Asia, South China, Hengduan Mountains, the northern part of Indochina, the southern part of Indochina, Malay Peninsula and South Asia (Fig. 2a). At the phylogenetic depth of 40 Ma, the northern and southern parts of Indochina and the Malay

Peninsula were merged into a united region. The southern part of the TP was separated from South Asia. The boundary between South China and North China & Korea moved from ca. 30 °N to 40 °N. At the phylogenetic depth of 20 Ma, the most striking change was that Central Asia combined with the Mongolian Plateau and emerged as an independent region. Then, Central Asia was separated from the Mongolian Plateau at a phylogenetic depth of 10 Ma. The spatial structures of the present-day zoogeographical regions are broadly similar to those at a phylogenetic depth of 5 Ma, when the whole TP was identified as an independent region. These findings are broadly consistent with the analyses performed on the whole-region species list for all terrestrial vertebrates (Supplementary Fig. 1). Notably, these biogeographical processes varied among taxonomic groups, as reflected by the spatial patterns (Supplementary Fig. 2) and the Mantel correlation test (Supplementary Fig. 3). The pβ$_{sim}$ structure between mammals and all terrestrial vertebrates showed the highest correlation in the present day (Supplementary Fig. 2 and 3). In contrast, their correlations were gradually weaker than those between ectotherms (i.e., reptiles and amphibians) and their combined counterparts in deeper phylogenetic time bins (Supplementary Fig. 3).

Temporal changes in the spatial structures of the zoogeographical regions were reflected in the UPGMA dendrograms and non-metric multidimensional scaling (NMDS) ordinations (Fig. 2b, c). Interestingly, the relationships between the zoogeographical regions over phylogenetic timescales were well illustrated by the Procrustes analysis (Fig. 3). For example, in the deep branches, the grid cells of North China & Korea and North Asia, and those of the Mongolian Plateau and Central Asia largely overlapped. In contrast, their differences were clearer in the shallow branches (Fig. 3). However, the relationships between West Asia and other regions underwent less change over phylogenetic timescales (Fig. 3). Interestingly, the species assemblage on the TP was similar to that of the Oriental realm in the past, but it became closer to the Palearctic realm towards the present day (arrow of the TP in Fig. 3 pointing to the Palearctic realm). This shift was also illustrated by the spatial patterns showing that the boundary between the TP and Mongolian Plateau became shallower in the present day, while the boundary between the TP and South Asia gradually strengthened (Fig. 2).

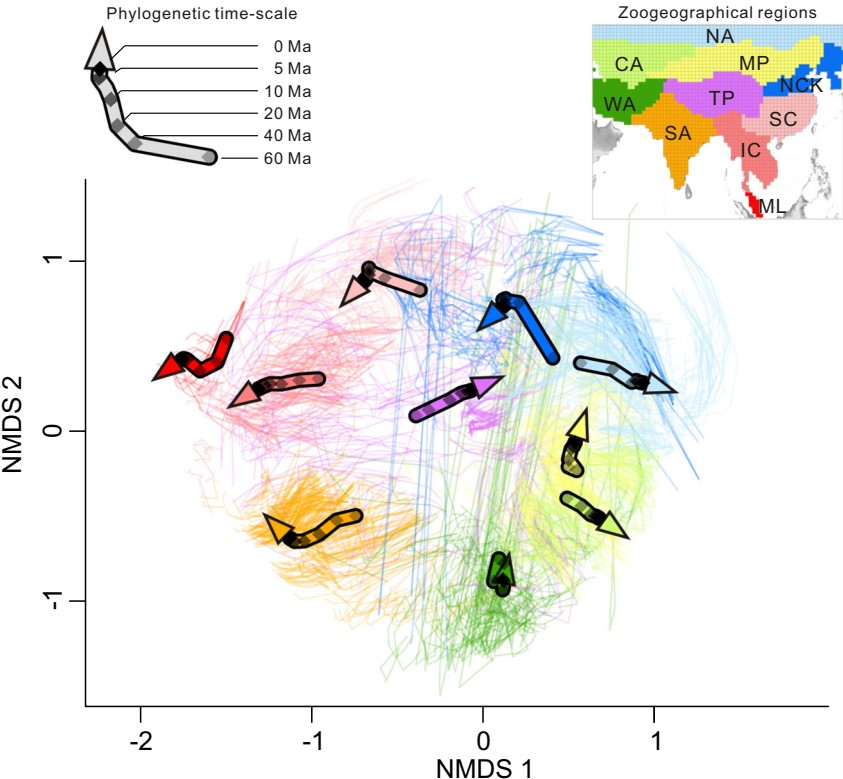

**Fig. 3 Temporal changes in terrestrial vertebrate communities in zoogeographical regions on a phylogenetic timescale.** Each gridded terrestrial vertebrate assemblage is represented by a small arrow linking the six coordinates through time (0 Ma, 5 Ma, 10 Ma, 20 Ma, 40 Ma and 60 Ma, respectively). Large arrows represent average gridded species assemblages across zoogeographical regions. Arrow colours correspond to the zoogeographical regions in the inset map. CA Central Asia, IC Indochina, ML Malay Peninsula, MP Mongolian Plateau, NA North Asia, NCK North China & Korea, SA South Asia, SC South China, TP Tibetan Plateau, WA West Asia.

**Zoogeographical regions over geological time.** The UPGMA clustering based on the fossil records showed that relationships between zoogeographical regions underwent a striking change through geological time (Fig. 4). In the Eocene (56–33.9 Ma), the TP was merged with South China and North China & Korea and then grouped with Central Asia and the Mongolian Plateau. South Asia emerged as the most distinct region, showing strong dissimilarities to the other regions (Fig. 4). During the Oligocene (33.9–23.0 Ma), after the initial collision between India and Eurasia, the TP was first grouped with South Asia. During the Early Miocene (23–15.9 Ma) and Mid–Late Miocene (15.9–5.3 Ma), however, the TP was first merged with the Mongolian Plateau. Then, the combination of the TP and Mongolian Plateau was grouped with the Oriental realm (i.e., South China and South Asia) in the Early Miocene, whereas they merged with the Palearctic realm (i.e., North China & Korea, North Asia and West Asia) in the Mid–Late Miocene. During the Pliocene–Pleistocene period (5.3 Ma–11.8 Ka), the division between the Palearctic and Oriental realms emerged. When we quantified $\beta_{sim}$ dissimilarity based on the extant mammal lists for the whole region, four groups of zoogeographical regions were identified, namely group 1: Central Asia + Mongolian Plateau + North Asia, group 2: South Asia + West Asia, group 3: Indochina + Malay Peninsula and group 4: North China + South China + Tibetan Plateau (Fig. 4).

**Comparison between phylogenetic and palaeontological inferences.** The faunistic relationships between zoogeographical regions based on phylogenetic information and fossil data yielded considerable differences. For instance, Central Asia combined

with the Mongolian Plateau emerged as an independent region at a phylogenetic depth of 20 Ma, while this pattern was not detected in the fossil data. The present-day zoogeographical regions broadly emerged at a phylogenetic depth of 5 Ma (Fig. 2; Supplementary Fig. 1), whereas the spatial structures of the zoogeographical regions between the Pliocene–Pleistocene period and present day displayed some differences based on fossil data (Fig. 4). Nevertheless, we found some consensuses among the phylogenetic and palaeontological inferences. For example, South Asia emerged as a distinct region in the early Cenozoic (compare Figs. 2 and 4). The faunistic similarity between the TP and Oriental realms was close in the early stages after the Indo-Asian Collision, whereas the TP has become more similar to the Palearctic realms since the Early Miocene (ca. 23–15.9 Ma, Figs. 2c–4).

**Discussion**

Our findings identified ten present-day zoogeographical regions surrounding the TP, which are broadly consistent with the regions identified by previous global regionalisation studies[5,6], despite different taxonomic groups being used as inputs. This indicates that different lineages share the similar ecological and historical drivers that underpin their co-occurrence distributions[8,34]. Notably, our results did not recognise the Sino-Japanese realm at a higher classification level as proposed by Holt et al.[5], as the present-day TP was first grouped with the Mongolian Plateau and then merged into other regions within the Palearctic realm (Fig. 2a, b). This pattern corroborated the assumption that the distinctiveness of the Sino-Japanese realm is rather weak and may be easily altered by slight changes in the

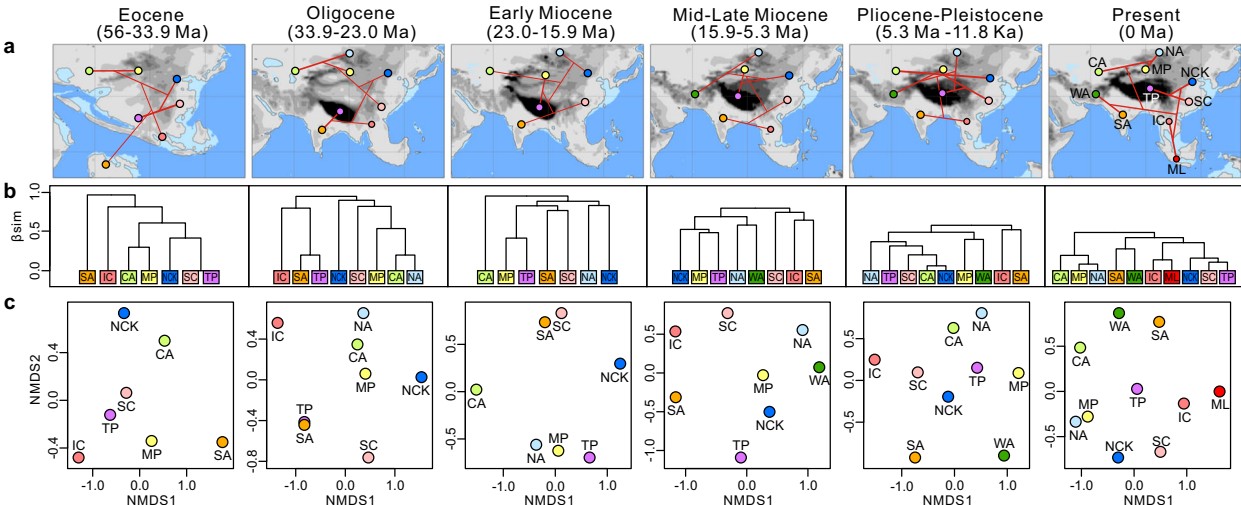

**Fig. 4 Temporal changes in mammal fossil assemblages between zoogeographical regions during the Cenozoic. a** Beta dissimilarities between zoogeographical regions inferred from UPGMA clustering are indicated by lines, with thinner lines indicating higher $\beta_{sim}$ dissimilarities. Palaeomaps were derived from a palaeo-digital elevation model developed by Scotese & Wright[52]. **b** Dendrograms from the unweighted pair-group method using arithmetic average hierarchical clustering of regional mammal lists during five time intervals and the present. **c** Non-metric multidimensional scaling ordinations based on the $\beta_{sim}$ matrices of regional mammal assemblages. CA Central Asia, IC Indochina, ML Malay Peninsula, MP Mongolian Plateau, NA North Asia, NCK North China & Korea, SA South Asia, SC South China, TP Tibetan Plateau, WA West Asia.

methods or taxonomic groups used[35]. Nevertheless, as this study focused only on the TP and its surrounding regions, further studies are required to assess the validity of the Sino-Japanese realm.

When we tracked the evolutionary history of the zoogeographical regions back to the early Cenozoic, South Asia (i.e., the Indian plate) was identified as an independent region based on both the phylogenetic and palaeontological results (Figs. 2 and 4). Although the phylogenetic distinctiveness of South Asia was lower than the palaeontological estimate, the independence of South Asia is still valid (Fig. 2). Tectonic studies have suggested that the Indian plate was part of Gondwana and was completely isolated from Eurasia before the Eocene continental collision (ca. 55–50 Ma[36]). As expected, it harbours several relict taxa closely related to the African and Madagascar lineages[37,38] that represent obviously distinct fauna from that of Eurasia. Notably, our results showed that the faunistic relationships between South Asia and the other zoogeographical regions within the Eurasian plate became closer towards the present day (Figs. 2 and 4). This shift is likely to reflect the imprint of the extensive biotic interchanges between South Asia and Eurasia since the Eocene[15,16]. Consequently, present-day South Asia shares a large proportion of extant lineages with Eurasia, and is placed in the Oriental realm instead of the African realm at a higher classification level[5,6].

It has long been suggested that the uplift of the TP reconfigured the spatial structures of the biota in Asia[7,14], but the faunistic relationships between the TP and its surrounding regions have not been adequately assessed. Our results revealed that the fauna on the TP was closer to the Oriental realm in deep time, but became more similar to the Palearctic realms towards the present time. This finding was independently confirmed by the p$\beta_{sim}$ dissimilarity of extant species (Figs. 2 and 3) and $\beta_{sim}$ dissimilarity of mammal fossils (Fig. 4). This trend is probably because, during the Eocene to Oligocene (ca. 56–23 Ma), the southern part of the TP began to emerge above sea level owing to the Indo-Asian Collision, whereas its northern part was still part of the Tethys Ocean[1]. Accordingly, some Palaeogene-aged lineages could have been expected to co-occur within both the Indian plate and TP[39]. However, as the uplift of the TP proceeded, the high, steep mountains in the southern TP began to act as a barrier

to biotic exchanges between the TP and the Oriental realm since the Middle Miocene (ca. 15.9–11.6 Ma[40]). In contrast, multiple species could disperse from the Palearctic realm into the TP via the northern routes due to the moderate topographic gradient[41,42]. Interestingly, only the southern part of the TP was identified at the phylogenetic depth of 40–10 Ma (Fig. 2), indicating that its species assemblage was more phylogenetically distinct in deep time. Both of these patterns provided biogeographical evidence that the TP underwent a south-to-north uplift process[7,43], and supported the recent palaeobotanical findings that the elevation of the southern part of the TP reached its present height at the Eocene–Oligocene boundary (ca. 34 Ma[44]).

We also found a profound change in that Central Asia combined with the Mongolian Plateau became an independent region at the phylogenetic depth of 20 Ma (Fig. 2). This timing is approximately contemporaneous with the aridification of the Asian interior during the early Miocene (ca. 24–22 Ma[45,46]), which was characterised by the contraction of the wet–humid biome and expansion of the dry–cool biome[47]. The aridification has been proposed to have created new habitats for speciation or dispersal barriers, and resulted in extensive vicariance events according to multiple taxon-specific studies[13,48]. Other evidence from mammalian fossils[49], palaeobotanical records[50] and sedimentary data[45] support such an important regional climatic change during this period and corroborate our phylogenetic results. However, the emergence of Central Asia and the Mongolian Plateau region was not observed in our fossil data (Fig. 4), which likely reflects the limitations of our palaeontological inferences (see details below).

Present-day spatial structures of the zoogeographical regions broadly emerged at the phylogenetic depth of 5 Ma (Fig. 2), as evidenced by the phylogenetic analyses based on both the gridded assemblages (Fig. 2) and whole-region species lists (Supplementary Fig. 1). According to the fossil data, although the spatial patterns during the Pliocene–Pleistocene period (5.3 Ma–11.8 Ka) were rather incongruent with those of the present day (Fig. 4), the correlation between their $\beta_{sim}$ dissimilarity matrices remained significant (Mantel test, $P < 0.05$; Supplementary Table 1). Previous palaeontological analyses in China yielded a similar timing and showed that the spatial structure of mammalian communities

originated during the Pliocene[9,51]. This probably happened because the emergence of modern orographic patterns[52] and monsoon systems[4] has formed dispersal limitations and accelerated lineage diversification. Many in situ radiation events within zoogeographical regions have been reported, including the pika *Ochotona* on the TP[53], the gibbon *Hylobates* in Indochina[54] and the palm squirrel *Funambulus* in South Asia[55], as well as several examples reported for other taxa (e.g., plants[56], fishes[57] and birds[58]). Although the subsequent Pleistocene glaciation cycle was expected to influence the geographical ranges of species and biodiversity distributions[59], our results detected negligible influences of this event on the spatial structure of the zoogeographical regions surrounding the TP (Fig. 2).

Notably, some discrepancies in the historical changes in zoogeographical regions emerged between taxonomic groups and analytical methods. For example, the ectotherms were more important than the endotherms in structuring the zoogeographical regions at deeper phylogenetic time bins (Supplementary Figs. 2 and 3). This probably results from differences in their life history strategies and evolutionary processes. The ectotherms are generally confined to fewer zoogeographical regions due to higher environmental sensitivity and weaker dispersal capacity[60]. Niche shifts in endotherms are faster than those in ectotherms[61], which have likely altered more $p\beta_{sim}$ patterns of endotherms towards the present[62]. Furthermore, the discrepancies in the phylogenetic and palaeontological inferences highlight the different outcomes of these methods. For example, the fauna of South Asia was clearly distinct from that of Eurasia during the early Cenozoic, but its phylogenetic dissimilarity was much lower than the palaeontological dissimilarity. This pattern illustrates that the past extinction of ancient endemism and geographical range shift caused by biotic interchanges might conceal considerable phylogenetic signals in deep time[63], and thus potentially biases the dissimilarity estimates[19,24]. Since the extant species represent only a subset of lineages in the phylogenetic tree, a comprehensive picture of the early history of a biota and its temporal changes cannot be resolved by the extant species alone[63]. However, analyses based on the fossil record fail to detect some biogeographical changes, such as the emergence of Central Asia and the Mongolian Plateau during the Early Miocene (ca. 23–15.9 Ma). One possible reason is that we had to merge the fossil records to coarse zoogeographical regions owing to the low number of fossils, which, however, only reflect the temporal changes in pairwise faunistic relationships between regions. Alternatively, the available fossil collection inevitably suffered from uneven sampling (Supplementary Table 2; see ref. [30]) and time averaging[31], making it impossible to clarify the finer-scale position of the zoogeographical boundaries and to reconstruct their successive temporal changes as the phylogenetic methods do[23]. Nonetheless, although there are some differences in the methods and inferred patterns between phylogenetic and palaeontological estimates, the comparisons between these two methods are still informative. For example, the changes in the relationships between the TP and its surrounding regions were effectively resolved by both methods. Overall, using community-level data to reconstruct the temporal changes in biogeographical regions is still a challenging and ongoing mission. Further studies are needed to integrate the molecular phylogenies and fossil data into a combined dataset[64], together with ancestral area estimates[65], to enable a more comprehensive understanding of evolutionary histories of present-day biogeographical patterns.

In conclusion, based on the long-term changes in beta dissimilarity inferred from the palaeontological data and phylogenetic information, this study reconstructed the evolutionary history of the zoogeographical regions surrounding the TP during the Cenozoic Era. Our study demonstrated that the faunistic relationships among these regions underwent a substantial reconfiguration during the Cenozoic as a consequence of several biogeographical events during different periods. These events included the Indo-Asian Collision, the TP uplift and the aridification of the Asian interior. The present-day zoogeographical regions surrounding the TP originated during the Miocene/Pliocene boundary (ca. 5 Ma) when the modern geographical pattern and climatic systems were established. The present study highlights the importance of comparing phylogenetic and palaeontological inferences to reconstruct the history of biogeographical regions. In this way, we may enhance our comprehension of the origin and evolution of life driven by various eco-evolutionary processes over space and time.

## Methods

**Species data.** We obtained extant species distribution maps from the IUCN Red List website (http://www.iucnredlist.org) for mammals and amphibians, Birdlife International and NatureServe (http://www.birdlife.org) for birds and Roll et al.[66] for reptiles. We excluded introduced, marine and domestic species. Species geographical ranges were transformed into presence and absence data in a matrix of 110-km × 110-km grid cells with the Behrmann projection. We removed grid cells with a land area <50% and species richness <5 to minimise the negative influences of the unequal sampling area and statistical uncertainty. We obtained the most comprehensive dated phylogenies available online (http://vertlife.org/phylosubsets) for each vertebrate group. For mammals, we used a phylogenetic tree (5911 species) from Upham et al.[67] that used two levels of Bayesian inference (backbone relationships and species-level phylogenies) to constrain the age and topological uncertainty. For birds, we used a phylogenetic tree (9993 species) from Jetz et al.[68] based on the Hackett family-level backbone. For reptiles and amphibians, we used phylogenetic trees from Tonini et al.[69] and Jetz & Pyron[70], comprising 9574 squamate species and 7238 amphibian species, respectively. In these phylogenies, the topology of species with molecular data was fixed, and the remaining species unsampled for DNA-sequence data were assigned randomly within their genus or higher-level groups based on morphology[69,70], resulting in a distribution of 10,000 trees. We downloaded a set of posterior distributions of trees ($n = 1000$) online using complete lists of all available species, and obtained the maximum clade-credibility phylogenies using the 'maxCladeCred' function from the 'phangorn' package[71] in R version 3.6.0[72]. After combining the distributional and phylogenetic data, our dataset comprised a total of 4966 extant terrestrial vertebrates, including 1022 mammals, 1741 birds, 1453 reptiles and 750 amphibians (Supplementary Data 1).

**Fossil data.** We obtained fossil records from four databases: Institute of Vertebrate Paleontology and Paleoanthropology, Beijing (http://www.ivpp.ac.cn/), the Paleobiology Database (https://www.paleobiodb.org/), the New and Old Worlds database (https://www.helsinki.fi/science/now/) and the Fossilworks database (https://fossilworks.org/), accessed in April 2018. We focused only on the mammal fossils owing to their relatively good preservation and samples[73]. In addition, we used genus rather than species as the analytical unit because the fossil records at the genus level included more complete sampling and reliable identification[64]. We standardised the taxonomy according to the Paleobiology Database and excluded taxa unidentifiable at the genus level. We removed the duplicated records and merged spatially closest collection localities by combining those within 0.1 latitude and longitude[9]. Our final fossil dataset consisted of 5880 fossil occurrences of 170 families and 1278 genera (Supplementary Data 2). We reconstructed the fossil records from present-day coordinates back to their palaeo-position based on the mean age of the fossil in a temporal range using the 'reconstruct' function in the 'chronosphere' package[74].

**Delineation of present-day zoogeographical regions.** To delineate present-day zoogeographical regions, we used Simpson's phylogenetic beta diversity ($p\beta_{sim}$) to generate pairwise dissimilarities between all pairs of grid cells using R package 'betapart'[75]. We calculated four $p\beta_{sim}$ matrices for individual taxonomic groups (mammals, birds, reptiles and amphibians) and generated combined $p\beta_{sim}$ matrices for all terrestrial vertebrates by taking the mean $p\beta_{sim}$ values[5]. We compared eight hierarchical clustering methods on the $p\beta_{sim}$ matrices and assessed the performance of different algorithms in transferring the dissimilarity matrices into dendrograms using cophenetic correlation coefficients[6]. As the UPGMA achieved significantly better performance than the other clustering algorithms (Supplementary Fig. 4), we only used UPGMA clustering for further analyses. We selected suitable cut-off points in the dendrograms using the 'recluster.region' function in the R package 'recluster'[76] based on the explained dissimilarity and mean silhouette width[5] considering the number of regions ranging from 2 to 15 (Supplementary Data 3). We defined the zoogeographical regions as the grid cells were geographically coherent and could be clearly delineated in space. We also ran NMDS ordination to investigate the relationships between zoogeographical regions based on the community compositions in two-dimensional space.

**Zoogeographical regions over phylogenetic time**. To assess the changes in zoogeographical regions surrounding the TP over phylogenetic timescales, we quantified pβ$_{sim}$ between gridded species assemblages at different phylogenetic depths[19,23]. This method cuts a phylogenetic tree at a specified depth and collapses all descendent leaves into ancestral branches[23]. When the geographical distributions of the descendent leaves were merged into their ancestral branches, a branch × site matrix emerged for a predefined depth. We cut the phylogenetic trees into different time bins from 60 Ma to 0 Ma and generated four pβ$_{sim}$ matrices for four individual taxonomic groups. We employed UPGMA clustering and NMDS ordinations to investigate the relationships among the gridded species assemblages based on the combined pβ$_{sim}$ matrices for four taxonomic groups in each time slice. Again, we used the explained dissimilarity and mean silhouette width to determine suitable cut-off points in the dendrograms. We investigated the evolutionary history of the zoogeographical regions based on the topological and assignment changes in the UPGMA clustering dendrogram and NMDS ordinations. In addition, we assessed the strength of the relationship between the present-day pβ$_{sim}$ matrices and those at different phylogenetic depths using the Mantel correlation test. To visualise the relationships between zoogeographical regions over phylogenetic time, we ran the NMDS for various time periods and maximised the correspondence between ordination pairs using Procrustes analysis[19] via the 'procrustes' function in the R package 'vegan'[77]. To assess the cross-taxon congruence in biogeographical processes, we performed these analyses separately for the four individual taxonomic groups.

**Zoogeographical regions over geological time**. To explore changes in the zoogeographical regions over geological time, we used Simpson's beta diversity (β$_{sim}$) to generate pairwise dissimilarities between fossil assemblages. Because sample completeness of the fossil records varies considerably in space and time (Supplementary Fig. 5), we assigned the fossil records to coarse spatial and temporal scales to strengthen the sampling intensity for each assemblage. To maximise the comparisons in the analyses based on the phylogenetic information, we assigned each fossil record to one of five time intervals: Eocene (56.0–33.9 Ma), Oligocene (33.9–23.0 Ma), Early Miocene (23.0–15.9 Ma), Mid–Late Miocene (15.9–5.3 Ma) and Pliocene–Pleistocene (5.33–11.8 Ka), and to one of the coarse-grained zoogeographical regions identified by the present-day phylogenetic dissimilarity. We performed UPGMA clustering analyses and NMDS ordinations based on the β$_{sim}$ matrices in different time intervals to explore changes in the zoogeographical regions over geological time. For present-day structures of the zoogeographical regions, we ran analyses based on the extant mammal lists for the whole region to maximise the comparisons of the fossil data.

**Statistics and reproducibility**. We used Wilcoxon signed-rank tests to compare eight hierarchical clustering methods on the β$_{sim}$ matrices. We used UPGMA clustering analyses and NMDS ordinations based on the β$_{sim}$ matrices in different time bins to explore changes in the zoogeographical regions over time. We used Mantel correlation tests to calculate the correlation coefficients of the β$_{sim}$ between each taxonomic group and all terrestrial vertebrates in different time slices. Statistical significance was calculated with a permutation test. A $P$ value of <0.05 was considered statistically significant. All statistical analyses were performed in R version 3.6.0[72]. All raw data and custom R codes are available from the Dryad Digital Repository (https://doi.org/10.5061/dryad.5x69p8d10[78]).

**Reporting summary**. Further information on research design is available in the Nature Research Reporting Summary linked to this article.

## Data availability

The species geographical ranges were based on the IUCN Red List database (http://www.iucnredlist.org), Birdlife International and NatureServe (http://www.birdlife.org), Global Biodiversity Information Facility (GBIF, http://www.gbif.org) and Roll et al.[66] (https://doi.org/10.5061/dryad.83s7k). The phylogenies for four vertebrate classes were available from the VertLife dataset online (http://vertlife.org/phylosubsets). The fossil data were compiled from the Institute of Vertebrate Paleontology and Paleoanthropology, Beijing (http://www.ivpp.ac.cn/), the Paleobiology Database (https://www.paleobiodb.org/), the New and Old Worlds database (https://www.helsinki.fi/science/now/) and the Fossilworks database (https://fossilworks.org/). The data supporting the findings of this study are available from Dryad Digital Repository (https://doi.org/10.5061/dryad.5x69p8d10[78]).

## Code availability

The R code used for this study is deposited in the Dryad Digital Repository (https://doi.org/10.5061/dryad.5x69p8d10[78]).

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

## Acknowledgements
We thank X. Liu for valuable discussions and comments that substantially improved this paper. This work was supported by grants from the National Natural Science Foundation of China (grant no. 31900324) and Guangdong Basic and Applied Basic Research Foundation (grant no. 2020A1515011472).

## Author contributions
J.H. and H.J. designed the study. J.H., S.L. and J.Y. collected the data. J.H., S.L., J.Y. and H.J. performed the analyses. J.H., J.L. and H.J. drafted the paper. All authors contributed to the final version of the paper.

## Competing interests
The authors declare no competing interests.
