## [Peer Review File · Communications Biology]

Editorial Note: *This manuscript has been previously reviewed at another Nature Research journal. This document only contains reviewer comments and rebuttal letters for versions considered at Communications Biology.*

REVIEWERS' COMMENTS:

Reviewer #1 (Remarks to the Author):

The authors combine present-day information on the distribution of vertebrates, phylogenetic information and data from fossils to improve knowledge of biogeographical history of Tibetan Plateau and the surrounding areas. The manuscript provides important insights on the biogeography of Asia. The major strengths of the manuscript include: the joint use of data from four clades of vertebrates, and the integration of present-day and fossil records.

I congratulate the authors for the integration of fossil data with their analysis. This is a very challenging task.

I only have a few comments.

L 69: a further disadvantage of this approach is that it only provides indirect evidence

L120-123: this is difficult to follow without checking Fig S3. It may be interesting adding Fig S3 to the main text. Do you have explanation for the fact that ectotherms are more important for the zoogeographical regions at the deepest phylogenetic time bins?

L249: do you refer to speciation or radiation events?

L270: I agree, but another problem is that the lower number of fossils forces you to a very coarse spatial resolution (e.g. one point per bioregion)

L322: the PBDB acronym is only used twice, I think it is unnecessary (just write Paleobiology Database at L 328)

Legend of Fig S2, L 732 : grid cells with species richness < 5

Reviewer #2 (Remarks to the Author):

The Authors present a revised manuscript enhanced by and expanded taxonomic scope and the additional examination of fossil mammal record to track the evolutionary history of regional assemblages surrounding the Tibetan plateau. This revision sees a significant expansion in the taxonomic scope of the study and makes use of extensive fossil mammal data. This has done much to address my previous concerns. I have only minor comments to add.

Minor Comments:

L71: There have been some studies that have had success with including fossils as tips in ancestral biogeographic analyses - (Dornburg et al., 2015; Siqueira et al., 2019).

L101: "When dating back in deep time" – this phrasing is a bit clumsy to read. Suggest "When tracing extant assemblages through deep time..." or more preferable – "At a phylogenetic depth of 60ma, nine clustered zoogeographic regions were identified".

L231: "new speciation habitat" does not read well – "new habitats for speciation" would be better.

L260: "dissimilarity in phylogenetic estimate" – would that not just be "phylogenetic dissimilarity"?

L261: "although still valid, was much lower." What make it "valid"? and "lower" compared to what? Some clarification need here.

L286: " and molecular phylogeny" – strictly speaking the phylogenies are not enitrie based on molecular data (see below). This need clarification

Methods/Species data:

I think more description is need here of how the phylogenetic data was constructed – from the VertLife website you can download sets of tree for different lists of taxa – how di you form the list of taxa" What was the criteria form inclusion the set?

You should give a brief note of how these trees were originall construction – I think all of them use some sort of polytomy resolution and species addition to add non-molecular sampled species to the trees – this should be noted.

Also it says you constructed a max clade credibility tree in phagorn (can you give the function and setting used if different from default) – how did you deal will low suppor values in tree? Given that all of the reconstructed trees use some sort of polytomy resolution/tip addition method this would lead to low node support in a max clade credibility tree? Were these low supported node collapsed prior to analyses?

Data Availability: I did not find a link to the Dryad repository for review of the code/data

Reviewer #1:

1. Comment: The authors combine present-day information on the distribution of vertebrates, phylogenetic information and data from fossils to improve knowledge of biogeographical history of Tibetan Plateau and the surrounding areas. The manuscript provides important insights on the biogeography of Asia.

The major strengths of the manuscript include: the joint use of data from four clades of vertebrates, and the integration of present-day and fossil records.

I congratulate the authors for the integration of fossil data with their analysis. This is a very challenging task. I only have a few comments.

Response: Thank you for the positive comments on our present work, and we appreciate your previous comments that substantially improved the earlier version of this manuscript.

2. Comment: L 69: a further disadvantage of this approach is that it only provides indirect evidence

Response: Thank you for this comment. We have added this disadvantage in the revised version (see Line 86-87).

3. Comment: L120-123: this is difficult to follow without checking Fig S3. It may be interesting adding Fig S3 to the main text. Do you have explanation for the fact that ectotherms are more important for the zoogeographical regions at the deepest phylogenetic time bins?

Response: Thanks for this suggestion, and we agree that it is rather difficult to compare the differences among taxonomic groups without checking Fig S3. However, we tend to remain this figure in the supplementary materials because our work focuses on the biogeographical patterns of all the terrestrial vertebrates, and presentation of this figure in the main text seems a bit inappropriate. In addition, we have added some explanations in the Discussion section for our finding that ectotherms are more important for the zoogeographical regions at the deepest

phylogenetic time bins (see Line 277-285).

4. Comment: L249: do you refer to speciation or radiation events?

Response: Thank you for this comment. We have written this sentence (see Line 269).

5. Comment: L270: I agree, but another problem is that the lower number of fossils forces you to a very coarse spatial resolution (e.g. one point per bioregion)

Response: We agree with the reviewer. We have rewritten this sentence (Line 298-300): “One possible reason is that we had to merge the fossil records to coarse zoogeographical regions owing to the low number of fossils, which, however, only reflect the temporal changes in pairwise faunistic relationships between regions.”

6. Comment: L322: the PBDB acronym is only used twice, I think it is unnecessary (just write Paleobiology Database at L 328)

Response: Done.

7. Comment: Legend of Fig S2, L 732 : grid cells with species richness < 5

Response: Done.

Reviewer #2:

1. Comment: The Authors present a revised manuscript enhanced by and expanded taxonomic scope and the additional examination of fossil mammal record to track the evolutionary history of regional assemblages surrounding the Tibetan plateau. This revision sees a significant expansion in the taxonomic scope of the study and makes use of extensive fossil mammal data. This has done much to address my previous concerns. I have only minor comments to add.

Response: Thank you for the positive comments on this version, and we are pleased

to have your approval.

2. Comment: L71: There have been some studies that have had success with including fossils as tips in ancestral biogeographic analyses - (Dornburg et al., 2015; Siqueira et al., 2019).

Response: Thank you very much for spotting the error and providing these papers on this issue. We have cited these papers and reworded this sentence in the revised version (Line 87-90): “Although the inclusion of ancestral range reconstruction in quantifying phylogenetic dissimilarity can improve estimates of evolutionary history, it is difficult to incorporate extinct lineages into the analysis (24, but see 28, 29).”

3. Comment: L101: “When dating back in deep time” – this phrasing is a bit clumpy to read. Suggest “When tracing extant assemblages through deep time...” or more preferable – “At a phylogenetic depth of 60ma, nine clustered zoogeographic regions were identified”.

Response: Done.

4. Comment: L231: “new speciation habitat” does not read well – “new habitats for speciation” would be better.

Response: Done.

5. Comment: L260: “dissimilarity in phylogenetic estimate” – would that not just be “phylogenetic dissimilarity”?

Response: Done.

6. Comment: L261: “although still valid, was much lower.” What make it “valid”? and “lower” compared to what? Some clarification need here.

Response: Thank you for spotting the sentence error. We have rewritten the sentence (Line 287-289): “For example, the fauna of South Asia was clearly distinct from that of Eurasia during the early Cenozoic, but their phylogenetic dissimilarity was much

lower than the palaeontological dissimilarity.”

7. Comment: L286: “ and molecular phylogeny” – strictly speaking the phylogenies are not entire based on molecular data (see below). This need clarification

Response: Thank you for spotting this error. We have rewritten the sentence “...inferred from the palaeontological data and molecular phylogenetic information, this...”

4. Comment: Methods/Species data: I think more description is need here of how the phylogenetic data was constructed – from the VertLife website you can download sets of tree for different lists of taxa – how did you form the list of taxa” What was the criteria form inclusion the set?

You should give a brief note of how these trees were original construction – I think all of them use some sort of polytomy resolution and species addition to add non-molecular sampled species to the trees – this should be noted.

Response: This is an important suggestion. We have provided more information on the phylogenetic trees and how we downloaded sets of tree with different lists of taxa (see Line 338-383).

5. Comment: Also it says you constructed a max clade credibility tree in phagorn (can you give the function and setting used if different from default) – how did you deal will low support values in tree? Given that all of the reconstructed trees use some sort of polytomy resolution/tip addition method this would lead to low node support in a max clade credibility tree? Were these low supported node collapsed prior to analyses?

Response: This is an important concern. Actually, we used the “maxCladeCred” function (by default) from the ‘phangorn’ package to generate a maximum clade credibility tree from a sample of trees ($n = 1,000$ in this study). It should be noted that, however, this function cannot yield a maximum clade credibility from a Bayesian posterior distribution. Instead, the “maxCladeCred” function results in a tree with the

highest clade credibility (i.e. the tree with the maximum sum of posterior clade probabilities) or a numeric vector of clade credibilities for each candidate tree, but it does not provide the support value of each node. As a result, our analyses were based on the whole max clade credibility tree and did not exclude any node prior to analyses.

4. Comment: Data Availability: I did not find a link to the Dryad repository for review of the code/data

Response: Thank you for this comment. We have provided the link to the Dryad repository in the revised version (see Line 457-459). This link will be available once this manuscript is formally published.

We appreciate for Editors/Reviewers' warm work earnestly, and hope that the correction will meet with approval.